# Adhesive Joint Integrity Monitoring Using the Full Spectral Response of Fiber Bragg Grating Sensors

**DOI:** 10.3390/polym13172954

**Published:** 2021-08-31

**Authors:** Chow-Shing Shin, Tzu-Chieh Lin

**Affiliations:** Department of Mechanical Engineering, National Taiwan University, No. 1, Sec. 4, Roosevelt Road, Taipei 10617, Taiwan; R04522534@ntu.edu.tw

**Keywords:** adhesive joint, integrity monitoring, fiber Bragg grating, tensile damage, fatigue damage, full spectral response

## Abstract

Although adhesive joining has many advantages over traditional joining techniques, their integrity is more difficult to examine and monitor. Serious structural failures might follow if adhesive joint degradation goes undetected. Available non-destructive examination (NDE) methods to detect defects are helpful in discovering defective joints during fabrication. For long-term monitoring of joint integrity, many of these NDE techniques are prohibitively expensive and time-consuming to carry out. Recently, fiber Bragg grating (FBG) sensors have been shown to be able to reflect strain in adhesive joints and offer an economical alternative for on-line monitoring. Most of the available works relied on the peak shifting phenomenon for sensing and studies on the use of full spectral responses for joint integrity monitoring are still lacking. Damage and disbonding inside an adhesive joint will give rise to non-uniform strain field that may chirp the FBG spectrum. It is reasoned that the full spectral responses may reveal the damage status inside the adhesive joints. In this work, FBGs are embedded in composite-to-composite single lap joints. Tensile and fatigue loading to joint failure have been applied, and the peak splitting and broadening of the full spectral responses from the embedded FBGs are shown to reflect the onset and development of damages. A parameter to quantify the change in the spectral responses has been proposed and independent assessment of the damage monitoring capability has been verified with post-damage fatigue tests.

## 1. Introduction

Adhesively-bonded joints are increasingly being used in aerospace, automotive and maritime industries. In contrast to traditional joining methods such as riveting, bolting and welding, adhesive joining does away with the hole drilling and localized heating to high temperature and keeps the structure surface smooth. Thus, there will not be material degradation in a heat affected zone nor structural weakening due to reduced cross-section and stress concentration. In composite materials, it helps to avoid fiber discontinuity. By spreading the load transfer throughout a large area, adhesive joints greatly reduce stress, leading to better stiffness and strength-to-weight ratios [1,2,3]. One disadvantage of adhesive joints is the difficulty in examining their integrity. Occasional overloading, long-term fluctuating service loading and adverse environment may damage and degrade these joints. If such degradation went undetected, serious structural failures and catastrophic outcome might follow. For example, the Aloha Airlines Flight 243 fuselage failure was believed to have originated from a degraded adhesive bond [4].

A number of non-destructive examination (NDE) techniques have been attempted to reveal defects in adhesive joints. Ultrasonic techniques based on a range of different working principles including the conventional pulse-echo [5,6,7] or through transmission [6] methods, and the more advanced guided wave technique [8,9], acoustic microscopy [10,11], and the electromagnetic acoustic transducer [12] have been proposed. Non-ultrasound techniques, such as electromechanical impedance spectroscopy using external [13] or embedded piezoelectric sensors [14,15], thermography [7,16,17] and shearography [6,18,19] have also been attempted. A number of the proposed techniques required localized point by point scanning or examination. For periodic inspection to ensure integrity, their applications in structures with large-scale adhesive joining will be very time-consuming and prohibitively expensive. Davis and McGregor [20] pointed out that most non-destructive inspection for defects techniques are mainly useful at the fabrication stage and at the late stage of failure but are ineffective during the joint degradation stage. Instead of directly looking for defects, there are relatively more economical techniques suitable for real-time monitoring of the integrity degradation of adhesive joints. These include strain/stiffness monitoring using back face strain gages [21,22,23], resistance monitoring of adhesive joints made conductive by adding carbon nanotubes [24,25] and optical fiber sensors signal surveillance [26,27,28,29,30,31,32,33,34,35,36,37,38,39,40,41,42,43]. Strain gages can only be applied to the outer surface, but they will disrupt an otherwise smooth surface and are susceptible to environmental degradation. They may not possess sufficient fatigue life for long-term monitoring [44]. The resistance method is economical to deploy to large adhesive joints but it is not easy to locate the damage sites. Its applicability to adherends with high resistivity may be limited. Optical fibers are known to have excellent fatigue endurance. They can be embedded inside the bond to leave the external surface smooth, which is important for aerodynamic structures. They are relatively free from environmental attack and have been used for general structural health monitoring [26,27,28]. There are different kinds of optical fiber sensors. In adhesive joints, both the distributed sensing [29,30,31] and the discrete fiber Bragg grating (FBG) sensors have been used [8,32,33,34,35,36,37,38,39,40,41,42,43]. The distributed sensors provide a spatially continuous strain measurement along the whole fiber and can reveal the location of any perturbation in strain caused by damages in the bond. With the discrete single peak FBG sensors, the peak wavelengths were often logged and converted to strain [32,33,34,35,36,37,38]. In this way they behave as strain gages but have advantages such as embeddability, multiple sensors on the same fiber and immunity from electromagnetic interference. However, automatic peak wavelength loggers will normally lock on to the peak with the highest intensity. If damages caused peak splitting and spectrum broadening, a single peak wavelength cannot reflect the actual strain status of the bond. By embedding a discrete chirp FBG sensor in the adhesive [39] or in one adherend close to the bondline [40,41], it has been shown that dip in intensity in part of the full spectral response occurred over artificially induced [39], or naturally initiated disbonds [40,41]. As the single peak FBG spectrum will become chirped if it is subjected to a non-uniform strain field, it is reasoned that if the full spectral response of a single peak FBG is used, information about the initiation and development of damages may be gained. Unfortunately, most of the single peak FBG investigations in adhesive joints treated the FBG as a strain gage. Applications making use of their full spectral responses are extremely limited [8,42,43]. Webb et al. [42,43] applied a dynamic full-spectrum interrogator to a single peak FBG sensor embedded in the adhesive of a single-lap joint. They detected change in the dynamic full spectral response caused by alternation of the strain field due to fatigue damage over a course of 600 cycles. Webb et al. [42,43] interrogated the full spectral response via an intensity modulated set-up [44] and extracted the peak wavelength information. This, in effect, used the FBG as a strain gage with very high frequency response rather than making use of the full information in the chirped spectrum. Karpenko et al. [8] attempted to use the information of the full chirped spectrum, but their adhesive joint had not yet been loaded to the point of damage initiation. The capability of the FBG spectrum to reveal damage is still not clearly known. In view of the above limitations, tensile and fatigue tests to failure will be carried out on adhesively-bonded single lap joint specimens. The capability and the method to make use of the full spectral responses of single peak FBGs to detect the onset and development of damages incurred during these tests will be investigated. Independent assessment using post-damage fatigue tests has verified the damage monitoring capability of the spectral responses.

## 2. Materials and Methods

### 2.1. Fiber Bragg Grating Sensor and Its Basic Properties

A fiber Bragg grating (FBG) is a certain section on an optical fiber with a periodic variation of refractive index. With a uniform period Λ, the grating will reflect a characteristic single peak spectrum with wavelength λ from an incident broadband light:λ = 2*n_e_*Λ(1)
where *n_e_* is the effective refractive index. When an FBG is subject to a strain, Λ will change and the corresponding reflected wavelength will shift accordingly. Temperature change will also change Λ as well as the refractive index, leading to peak wavelength shift. In general, the reflected spectrum will shift by the order of ~1 pm under a uniform strain of 1 με, and ~10 pm per degree Celsius change. In this work, mechanical tests were carried out in an air-conditioned room with thermostat control to ±1 °C and the FBGs were embedded in poor thermal conductors of polymeric adhesive sandwiched between composite laminates so that it is relatively insensitive to outside temperature changes. Small ambient temperature fluctuations will have negligible effect on the measured spectra. In applications where temperature variations are significant, its effect on the spectrum must be taken into account during the interpretation of results.

Figure 1 schematically shows the stress distributions along the centerline of an adhesive joint under tension and the possible resulting effects on the FBG spectrum. First of all, the average stress along the fiber (*σ_xx_*) is tensile and this will cause elongation of the grating period Λ. From equation 1, an increase in Λ will shift the whole spectrum towards the longer wavelength. Secondly, the varying stress field, from slight compression to large tension, perturbs the uniformity of the grating period. Thus Λ and as a result λ are not unique single values. A series of wavelengths instead of a single wavelength may then satisfy Equation (1), leading to a broadening or chirping of the original narrow single peak spectrum. Thirdly, the high stresses near the longitudinal edges (*x* = 0 and 12.5 mm) stretches that portion of FBG heavily and may lead to secondary peaks arising on the long wavelength end, Finally, the existence of transverse stresses (*σ_yy_* and *σ_zz_*), if large enough, may split the single peak into two [45]. In practice, each of the above phenomena occurs to different degrees, according to the magnitude and the extent of the stresses involved, and superimpose together to give a resulting spectrum. In damage monitoring using embedded FBGs, further complications will arise as embedment incurs residual stresses and internal damages change the stress distribution.

In this work, single-peak FBGs were used and were fabricated in a Ge–B co-doped single mode optical fiber by side writing using a phase mask. The sensing length of the FBGs was about 10 mm. The reflectivity of the as produced FBG was about 99%. The reflected spectra from the FBGs were interrogated using an optical spectrum analyzer (MS9710C, anritsu, Kanagawa, Japan).

### 2.2. Single Lap Joint Specimens

Strips with dimensions of 101.6 mm × 25.4 mm were cut from a 220 mm × 220 mm graphite-epoxy composite laminate consisting of 10 uni-directional plies. Each two of these strips were glued together with Loctite E-30CL structural epoxy adhesive (Henkel Taiwan Ltd., New Taipei City, Taiwan) to form single lap joint specimens. The fiber direction in the composite was along the loading axis. The areas to be joined were sanded and masking tape was applied to the immediate vicinity beyond the boundary of the joint area. The purpose of the tape was to prevent excess glue resulting in additional, but unpredictable adhesion between the two parts. Excess glue was difficult to clean away, especially with the optical fibers in place. Three optical fibers with FBGs were embedded in the joint. The bond line was approximately as thick as the diameter of the optical fiber, i.e., 125 μm. The sensors were designated as FBGL, FBGM and FBGR to indicate whether they were near the left side edge, in the middle or near the right-side edge. A section of the same composite strip was also glued to each end of the specimen to ensure the loading axis passed through the center of the adhesive layer. Detailed dimensions and layout of the specimens are shown in Figure 2.

### 2.3. Mechanical Testing

A batch of seven single lap joint specimens can be made from a 220 mm × 220 mm composite laminate. Preliminary test showed that their tensile strengths are affected by environmental conditions such as temperature and humidity during the joining operation. Different specimen batches prepared over a course of two years had average batch strengths from 5.16 to 7.81 kN. However, within the same batch, the worst-case standard deviation of tensile strength was within 6%. Thus, for each batch of 7 specimens, 2 randomly sampled specimens were tested under monotonic loading to obtain the average batch tensile strength while the remaining specimens were used for various tests and measurements. The specimens were subjected to tensile or cyclic loading on a servo-hydraulic testing machine (810 Materials Testing System, MTS Systems, MN, USA).

During the tensile test, loading was periodically interrupted to allow the reflected light spectra from the FBGs to be recorded at the instantaneous loading. The specimen was then unloaded to allow reflected spectra to be measured at reference loads of 0 N and 400 N. This loading–unloading cycle was repeated with progressively higher loading until specimen failure.

Fatigue testing was carried out with a cyclic loading range of 4.5–45% of the average batch tensile strength at 8 Hz. Again, the tests were interrupted periodically to allow the FBG spectra to be measured at the reference loads. Three types of fatigue tests were carried out: (i) virgin specimens were subjected to cyclic loading until failure; (ii) cyclic loading was applied after specimens were pre-loaded to somewhere above 45% of the average batch tensile strength while no significant change to the FBG spectrum was apparent; (iii) cyclic loading was applied after specimens were tensile pre-loaded to the extent that noticeable change to the FBG spectrum started to appear.

For each set of testing conditions, the tests were repeated at least five times to make sure that the results are consistently reproducible.

## 3. Results

### 3.1. Damage Monitoring during Tensile Tests

#### 3.1.1. Spectrum Evolution under Tensile Loading

Figure 3 shows the spectra from the three FBGs in a typical tensile specimen under different loading. As the applied tensile loading increased from 0 to 3200 N, the spectra progressively shifted towards longer wavelengths. The FBGL and FBGR spectra basically retained their shapes while the FBGM spectrum exhibited some enhanced secondary peaks. At 5200 N, beside shifting further to the right, secondary peaks in FBGL started to enhance while that of FBGM and FBGR intensified significantly so that the spectra have markedly broadened or chirped. Broadening became very prominent in all three spectra at 6600 N. The background intensity also rose significantly in the FBGM and FBGR spectra. This specimen eventually failed at 6797 N.

Exact reason for the change in background intensity is not known. Most of the times, the change aggravates with damages. The same phenomenon has also been observed in impact and fatigue damage-monitoring, where the background intensity rose to such extent that the reflected peak disappeared [46]. Improper connections, variation in source intensity, twisting and straining of the free lead fiber beyond the embedded portion have been precluded from the cause. It was shown that the narrow single peaked spectrum was restored when the embedded FBG was carefully extracted from the damaged specimen [46]. This background intensity change is probably caused by the straining of the embedded portion of the fiber. As noted above in Figure 1, the reflected spectrum from an FBG depends on the grating period, which is affected by the strain acting on the FBG. Broadening and chirping of the spectra is caused by a development of non-uniform strain distribution along the FBGs. Stress analysis of single lap joint [47] indicated that for each of the stress components, stress concentration occurs right at or very close to the longitudinal ends of the joint (PQ and RS in Figure 2). Both the stress concentration near the edges of the joint [47] and the development of internal damage can give rise to non-uniform strains. These two effects will be aggravated by increasing load, giving rise to the observed evolution of the spectra in Figure 3. Obviously for damage-monitoring, the effect of loading must be precluded. To this end, after loading to different levels, the specimens were unloaded to some small reference loads for the spectra to be measured.

#### 3.1.2. Damage Monitoring Using Unload Spectra

Two reference loads, namely 0 N and 400 N, were chosen. In each case, as measurements were made under the same loading, the effect of loading would either be absent or stay the same unless the condition inside the joint was changed by damages. At zero load, small early damages may remain shut and exhibit minimal perturbation on the strain distribution. The higher load of 400 N was intended to open up small damages and amplify the effect of damage on the spectra. Figure 4 shows the unload spectra measured at 0 N from the same specimen as above. The 3200 N unload spectra virtually overlapped with the reference spectra recorded initially at 0 N at the beginning of the tensile test, suggesting negligible damage has arisen at 3200 N. The 5200 N unload spectra exhibited slight deviation from the reference, indicating that damage has probably commenced. The 6600 N unload spectra were heavily chirped, showing that damage has probably became extensive. The reason that spectra at zero load will change with the occurrence of damage may be attributed to residual stress inside the adhesive joint that invariably exist. Occurrence of damages perturbs the residual stress field, leading to changes in the spectral responses.

Unload spectra measured at 400 N are shown in Figure 5. The development of the spectra is basically the same as that under 0 N. Raising the reference load to 400 N has no discernible difference between the spectra unloaded from 3200 N and the reference. When compared with those in Figure 4, it does slightly enhance the differences between the 5200 N unload spectra and the respective reference spectra for FBGM and FBGR. Its enhancement effect on FBGL spectra is not marked. The above observation agrees with the reasoning that negligible damage at 3200 N and existence of damage at 5200 N. It further suggests that the initial damage near FBGL is less serious than that near FBGM and FBGR.

Fracture surface observation using the scanning electron microscope shows the joint typically failed by brittle cohesive failure of the adhesive and intralaminar failure of the adherend very close to the bond line (Figure 6a). A higher magnification view (Figure 6b) shows more clearly that the bond did not fail at the adhesive–adherend interface but inside the adherend lamina. Prominent plastic deformation marking in the resin can be seen in Figure 6b. Since brittle fracture occurs with little deformation, it can be reasoned that cracking of adhesive occurred first and plastic deformation of intralaminar resin and its final failure took place at a later stage. Cracking of the adhesive will perturb the original residual stress distribution after curing and lead to changes in the unload spectra observed in the early stage of damage in Figure 4, Figure 5 and Figure 7 shows the remnant of the optical fiber on the fracture surface. The glue adhered well to the fiber and showed some brittle cohesive failures while the lips sticking out of the fracture surface indicate marked plastic deformation. As noted above, the plastic deformation probably occurred at the late stage near the final fracture. These fractures and deformation in the adhesive, as well as those that took place in the adherend resin, contributed to the changes in the unload spectra observed in Figure 4 and Figure 5.

#### 3.1.3. Quantitative Comparison of Unload Spectra with the Reference Spectrum

To discern the occurrence of internal damage, one has to compare the instantaneous unload spectra with the reference. Visual comparison involves subjective judgement and is qualitative only, both are not preferable in practical applications. To aid objective and quantitative assessment, a parameter *V* is defined:(2)$$V=\frac{1000\sqrt{\sum_{i=1}^{n}{\left(P_{\lambda_{i}}^{current}-P_{\lambda_{i}}^{reference}\right)}^{2}}}{\sum_{i=1}^{n}P_{\lambda_{i}}^{reference}}$$
where $P_{\lambda_{i}}^{current}$ and $P_{\lambda_{i}}^{reference}$ are, respectively, the power intensity in dbm of the current unload spectra and the reference spectra at the same corresponding wavelength *λ_i_*. *n* is the number of data points in the wavelength span chosen to characterize the reference as well as the subsequently shifted and deformed spectra.

Figure 8a,b shows the development of *V* values from the unloaded spectra at 0 N and 400 N, respectively. The *V* values increased gradually with increasing load and the *V* from the three FBGs essentially fall on the same scatter band initially. At about 70% failure load, which is about 4.8 kN, the rate of increase in *V* became steeper in the FBGM and FBGR spectra. Those from FBGL started to gain speed at about 5.2 kN. The turning point corresponds to a *V* value of ~10, which may be used to judge the starting point of development of significant damage. The evolution of *V* in Figure 8a suggests damage development is slow before 4.8 kN. In the region near FBGM and FBGR, damage developed quickly after 4.8 kN while near FBGL, damage development accelerated after 5.2 kN. Note that visual comparison of the spectra gives nearly no discernible change before 3200 N in Figure 4 and Figure 5 above, but the *V* values showed slow yet steady increases. This suggests that the *V* value is more sensitive in reflecting the change in the spectra. Figure 8b, derived from spectra measured at a reference load of 400 N, shows essentially the same trends as Figure 8a. The higher reference load of 400 N does not seem to enhance either the spectra deviation or the *V* value to allow damages to be revealed earlier. In later sections, only the 0 N load will be used for reference.

In Figure 8b, instead of increasing monotonically, the *V* for FBGM drops before rising again near the end of the test. A possible explanation of this phenomenon may rest on two antagonistic effects as follows: (1) cracks resulting from brittle fracture of the adhesive (Figure 6a) at the early stage of damage will act as stress raisers, increasing the local residual tensile stress and shifting part of the FBG spectrum to longer wavelength, away from the reference spectrum. This results in an increasing *V*. (2) At the late stage of damage, plastic deformation in the resin (Figure 6b) may cause local disbonding, relieving stress from part of the FBG. This will tend to move the spectrum to the shorter wavelength, causing more overlap with the reference spectrum and decreasing *V*. If the latter effect dominates, a drop in *V* will result. This is a limitation of *V* that does not increase monotonically with the degree of damage when more violent damage behavior occurs.

In our work, temperature fluctuation was insignificant, as noted above. In practical applications, temperature variation may not be negligible. The total *V* value will then comprise of components due to damage and temperature change respectively. In this case, the effect of temperature on the reference spectrum should be calibrated beforehand and *V* should be computed against a reference spectrum at the measuring temperature to reflect the net contribution of damages.

### 3.2. Damage Monitoring during Cyclic Fatigue Loading

#### 3.2.1. Virgin Specimens Fatigue Testing

Seven specimens from three different batches were tested under a cyclic load range between 4.5% to 45% of their average batch failure strengths. The fatigue lives and the corresponding batch strength of individual specimens are listed in Table 1. The average fatigue life is 184,200 cycles, with a standard deviation of 43,273 cycles. Figure 9 shows the evolution of the unload FBG spectra during the course of fatigue life from fatigue specimen 1. For FBGM and FBGR, the change in the spectra was very gradual up to 120,000 cycles, or ~71.5% of life. There were only slight shifts in peak wavelength and appearance of weakly discernible secondary peaks. These kinds of developments maintained till the last spectra, taken at 165,000 cycles (~98.2% of fatigue life). For FBGL, besides the above pattern of changes, the background intensity also rose. As noted before, the exact mechanism that causes the rising background intensity is not clear but the phenomenon is related to aggravating internal damages. Evolution in the *V* values (Figure 10) indicates the change was slow before ~50% of life and it mildly accelerated afterward. Figure 10 shows that *V* for FBGM is not monotonically increasing after ~70% of fatigue life. The reason may be the occurrence of complex interactions of damages close to that FBG as explained in the last section.

All in all, the expression of damage in the FBG spectra is much more gentle under fatigue damage at this loading range than that in tensile failure. In tensile test, at loading above a certain percentage of the tensile strength, the large average stress coupled with the stress concentration effects of the joint and the defects will cause widespread damage, giving rise to very marked change in the FBG spectra considerably before the final failure. On the other hand, fatigue damage is highly localized and stochastic in nature. Under small cyclic loading, initial damages develop locally and gradually. Extensive spread to all over the joint may probably occur extremely close to the end of life. Periodic recording at an increment of a few thousand cycles may easily miss the final stages of failure. Ff such initial damages happen to not occur in the close vicinity of an FBG, the change in the spectra will not be marked throughout most of the life duration. These explain the relatively gentle changes in the spectral responses observed in Figure 9. Again, quantification the change in the spectra with the *V* values helps to visualize the change and the degree of damage more clearly.

#### 3.2.2. Fatigue Testing of Specimens Pre-Tensioned to before Damage Emergence

Tensile loading was applied to a batch of six specimens which were subjected to stepwise increasing loading to above the maximum cyclic load but just stopped short of causing discernible change in the unload spectra. Figure 11 shows the spectra of a typical specimen loaded to and unloaded from a maximum load of 4100 N, which corresponds to about 51.3% of the average batch tensile failure strength. Besides a slight increase in intensity on the long wavelength portion of the peaks, the spectra exhibited no sign that characterize the occurrence of damage such as peak shifting, splitting, emergence of secondary peaks, and chirping. The maximum *V* values of the three spectra are all well below 10 (Figure 12), which is the turning point value between slow gradual increase and steep increase as established in Figure 8. On fatigue cycled between 4.5% and 45% of tensile strength, it failed after 134,900 cycles. This is within the scatter of the average virgin specimen fatigue lives. As in the virgin specimens, the unload spectra of FBGL and FBGM only showed very slight changes up to 130,000 cycles, or 96.4% of total life (Figure 13). The FBGR spectra, while maintaining an unsplit single peak, showed a gradually rising background intensity.

Testing of six specimens in this way gave fatigue lives ranging from 122,400 to 166,700 cycles, all within the scatter of virgin specimen fatigue life. This suggests the absence of spectrum-splitting, emergence of secondary peaks, chirping and peak shifting is associated with the absence of significant damage in the adhesive joint.

#### 3.2.3. Fatigue Testing of Specimens Pre-Tensioned to Damage Emergence

In this batch of five specimens, tensile loading was increased stepwise to just causing discernible deviation of the unload spectra from their references. Figure 14 shows the spectra of a typical specimen loaded to and unloaded from a maximum load of 3600 N, which corresponds to about 52% of its average batch tensile failure strength. Emergence of secondary peaks, though weakly discernible, is evident in all three spectra. Slight shifting of the peak wavelength can also be seen. The *V* values of the three FBG spectra are all above 10 (Figure 15). By the standard established above in Figure 8, significant damage has just started to appear when *V* > 10. On fatigue cycled between 4.5% and 45% of tensile strength, the specimen failed after 38,200 cycles. Of the five specimens tested in this way, fatigue lives ranged from 14,000 to 44,700 cycles. All these are significantly lower than that of the average virgin specimens. The unload spectra during cyclic loading deviated markedly from their respective references (Figure 16). In contrast to fatigue of specimens with no pre-existing damage which showed gradual and very slight change at above 95% of their lives, splitting, broadening and appearance of secondary peaks can clearly be seen in the spectra shown in Figure 16 at 25,000 cycles, or ~65% of fatigue life.

Summarizing the above two types of pre-tensioned fatigue results, one may conclude that the appearance of secondary peaks, peak splitting, shifting and spectrum broadening of the unload spectra signify the onset of damage and aggravation of the above spectrum phenomena is associated with development of the damages.

## 4. Conclusions

Development of the full spectral responses of single-peak FBG sensors embedded in composite single lap adhesive joints has been studied under tensile and fatigue failure of the joints. It has been demonstrated that the load-induced damages inside the joint are expressed as peak shifting, peak splitting, emergence of secondary peaks and broadening of the FBG spectra. To successfully monitor the initiation and development of joint damages, the response from spectra measured at a fixed and low reference load should be used. In this way any effect of deformation and stress concentration due to loading will be excluded. A quantitative parameter *V* has been proposed and shown to sensitively and objectively reflect the difference between the instantaneous unload spectrum and the reference spectrum. It has a good potential to develop into an objective parameter for damage monitoring using FBG sensors. In practical applications, for sensitive detection of internal bond damages, it is recommended to embed the optical fiber in the direction of loading so that the FBG portion covers the most severely stressed region. The full spectral responses should be recorded under the load-free condition or at a fixed low reference load. Judgement of the degree of damage may then be made by directly compare the spectra with the reference, or indirectly through the quantitative parameter *V*. Temperature at the FBG should also be recorded and the temperature effect on the reference spectrum should be calibrated first. If the unload spectra are measured at a different temperature from the reference, the latter should be compensated for the temperature variation before making a comparison.

## Figures and Tables

**Figure 1 polymers-13-02954-f001:**
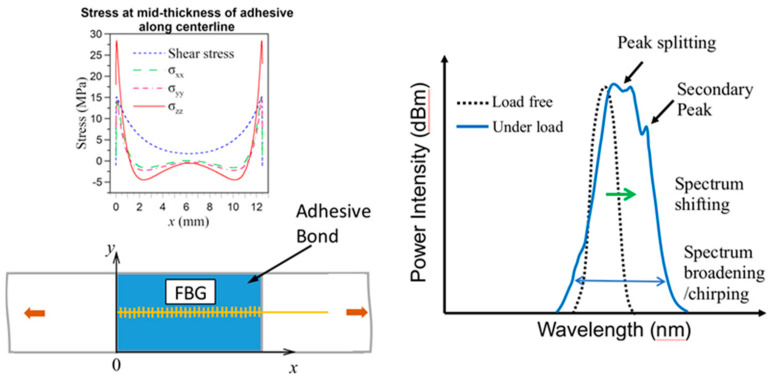
Schematic diagram showing the stress distribution along the centerline of the adhesive joint and its possible resulting effects on an FBG embedded at that position.

**Figure 2 polymers-13-02954-f002:**
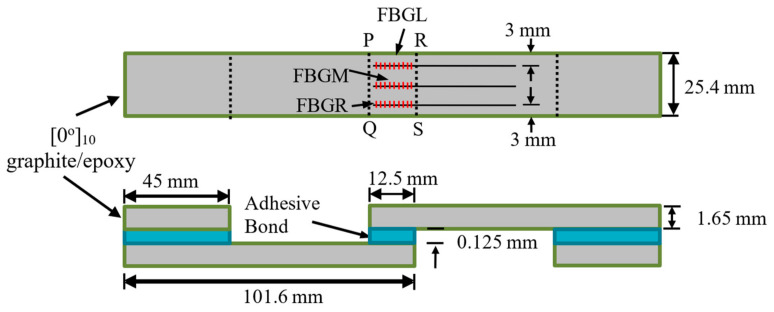
Dimensions and layout of the single lap joint specimen.

**Figure 3 polymers-13-02954-f003:**
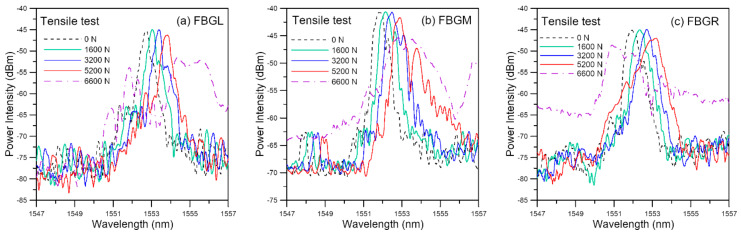
Evolution of FBG spectra measured at various loads as tensile loading increases.

**Figure 4 polymers-13-02954-f004:**
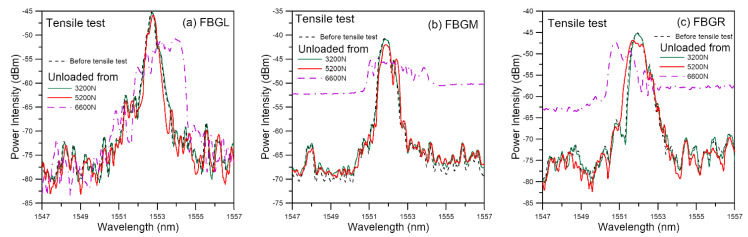
FBG spectra measured at 0 N after unloading from various tensile loads.

**Figure 5 polymers-13-02954-f005:**
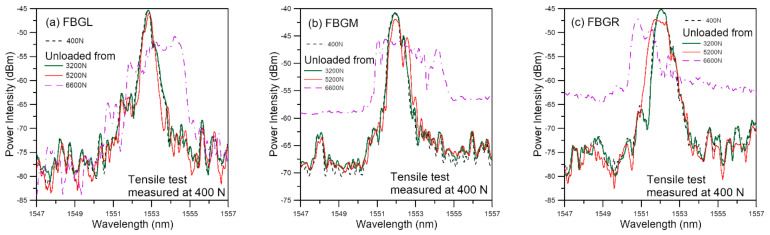
FBG spectra measured at 400 N after unloading from various tensile loads.

**Figure 6 polymers-13-02954-f006:**
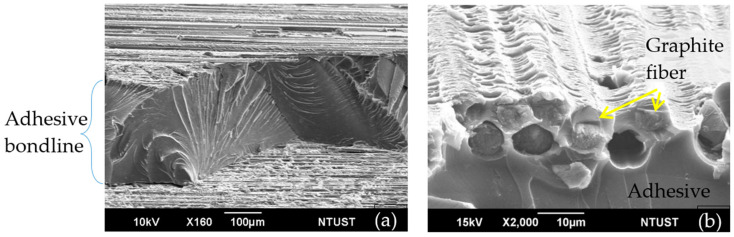
Scanning electron micrograph of a typical fractured joint. (**a**) Brittle cohesive fracture of adhesive; (**b**) intralamina failure of the adherend with plastic deformation markings in the resin.

**Figure 7 polymers-13-02954-f007:**
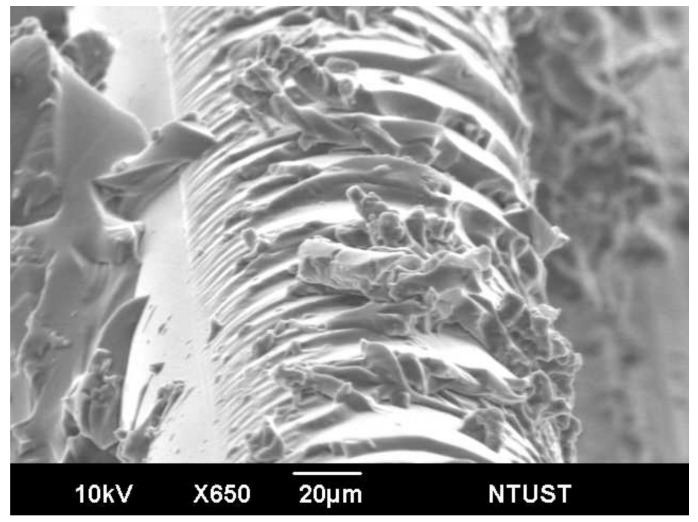
Scanning electron micrograph showing the embedded optical fiber on the fractured joint.

**Figure 8 polymers-13-02954-f008:**
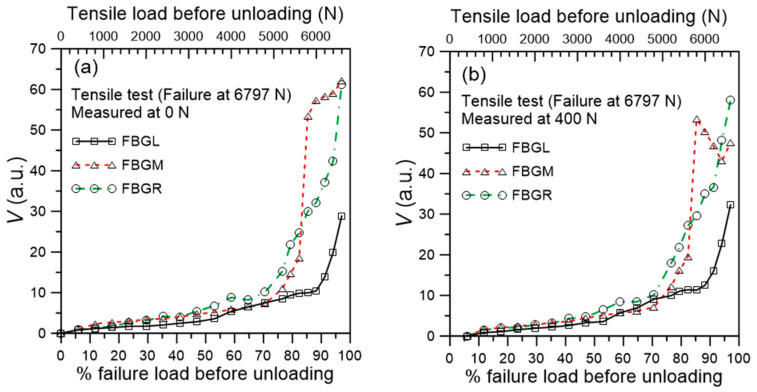
Evolution of the *V* values of FBG spectra unloaded from various tensile loading measured at: (**a**) 0 N; (**b**) 400 N.

**Figure 9 polymers-13-02954-f009:**
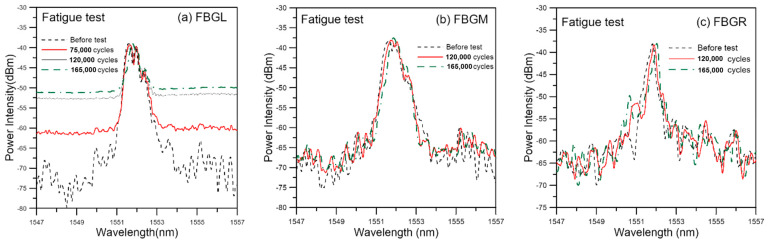
Evolution of unload FBG spectra during fatigue cycling of a virgin specimen.

**Figure 10 polymers-13-02954-f010:**
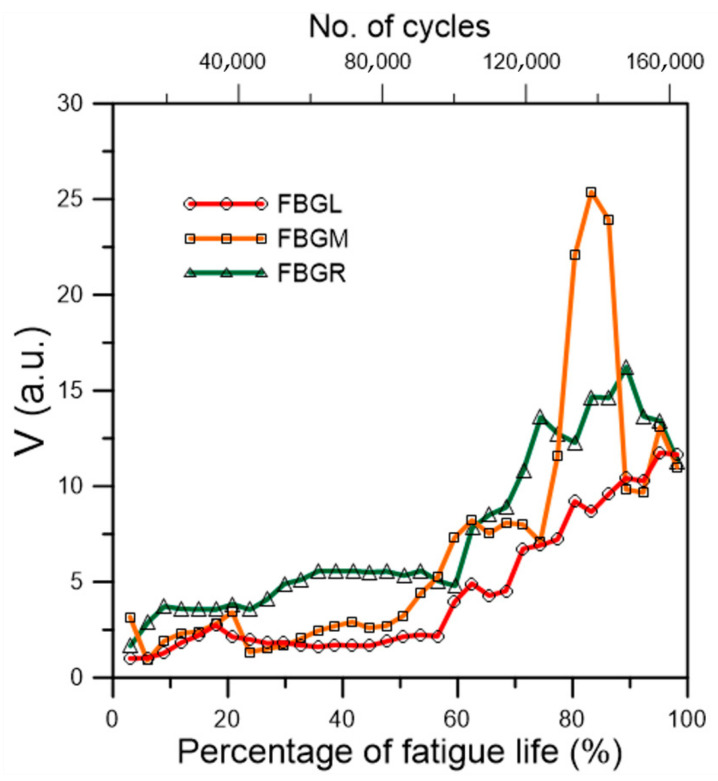
Evolution of the *V* values of the three FBG unload spectra during fatigue cycling of a virgin specimen.

**Figure 11 polymers-13-02954-f011:**
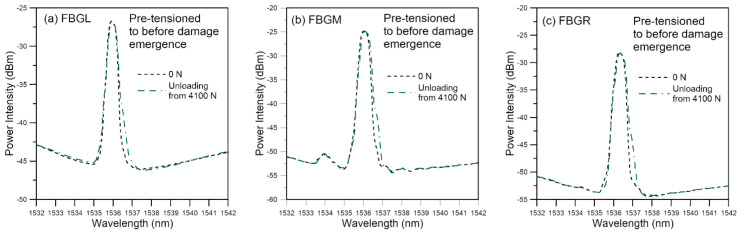
Unload spectra of fatigue specimen pre-tensioned to before damage emergence.

**Figure 12 polymers-13-02954-f012:**
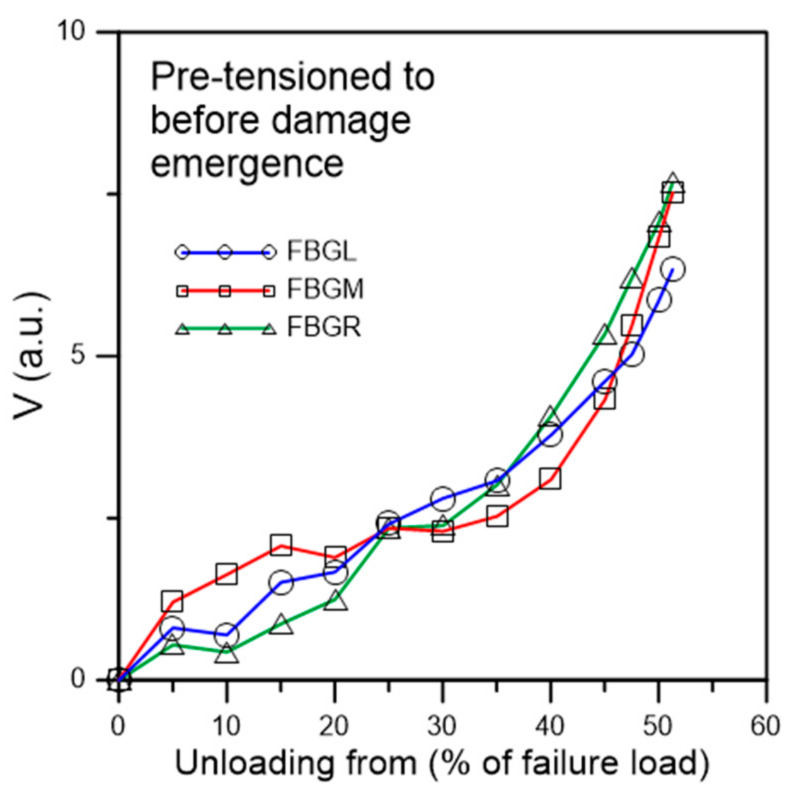
*V* value from unload spectra of fatigue specimen pre-tensioned to before damage emergence.

**Figure 13 polymers-13-02954-f013:**
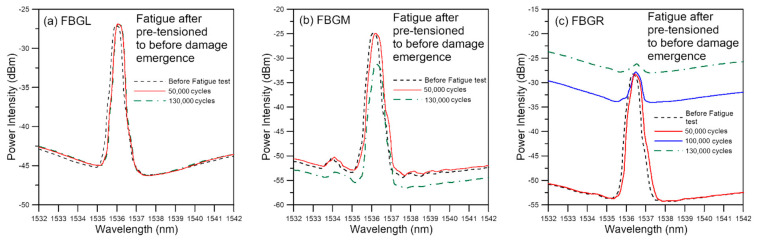
Evolution of unload FBG spectra during fatigue cycling of a specimen pre-tensioned to before damage emergence (failure at 134,900 cycles).

**Figure 14 polymers-13-02954-f014:**
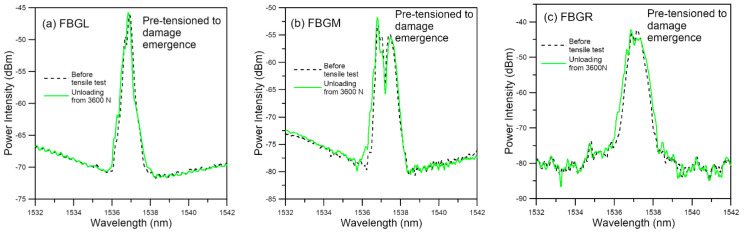
Unload spectra of fatigue specimen pre-tensioned to damage emergence.

**Figure 15 polymers-13-02954-f015:**
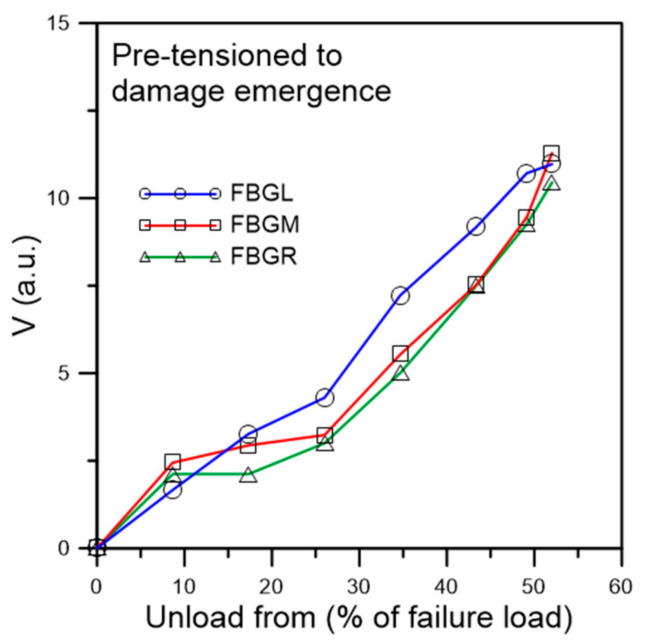
*V* value from unload spectra of fatigue specimen pre-tensioned to damage emergence.

**Figure 16 polymers-13-02954-f016:**
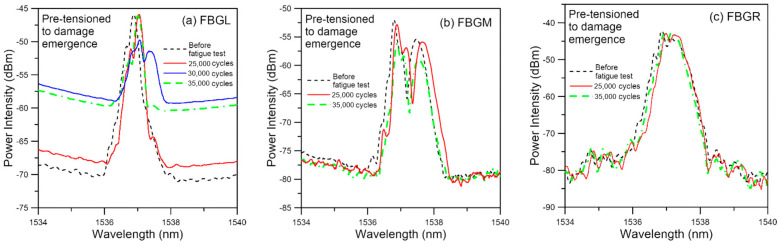
Evolution of unload FBG spectra during fatigue cycling of a specimen pre-tensioned to damage emergence.

**Table 1 polymers-13-02954-t001:** Fatigue lives and the average batch tensile strength of virgin specimens tested.

Specimens	Fatigue Life (Cycles)	Batch Tensile Strength (kN)
1	168,014	5.74 ± 0.11
2	265,800	5.47 ± 0.2
3	227,300	
4	157,670	
5	175,606	5.49 ± 0.17
6	168,229	
7	126,778	

## Data Availability

The data presented in this study are available on request from the corresponding author.

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
