# Peer review of "Adhesive Joint Integrity Monitoring Using the Full Spectral Response of Fiber Bragg Grating Sensors"

_polymers, 2021, doi:10.3390/polym13172954_

Round 1

Reviewer 1 Report

Authors presented the analysis of FBGs under tensile loads. The goal is to evaluate the structural health of adhesive joints, which would result in distortions (chirp) in the gratings spectra. Fatigue tests were also performed. In general, the paper is important for the sensor's community as it provides interesting analysis on sensor's responses and applications. However, the paper needs a few corrections before publication.

  • English revision is needed.
  • Authors should revise and improve the introduction. Although it includes some literature background, a discussion about the paper contributions should be included.
  • If such large number of cycles, the temperature stability as well as the sensor temperature sensitivity should be discussed. Specially when the fiber is embedded in another material, which influences its temperature sensitivity.
  • This reviewer misses a deeper discussion on the reasons for the chirp effect on the fiber considering each performed experiment.
  • Minor correction: always include the unit of the load, there are some parts where the force has no unit, e.g., line 195, line 197.
  • Authors should provide a broader discussion if the parameter V can be used as failure indicator in adhesive joints, including in which conditions it could be analyzed.

Reviewer 2 Report

Dear Author,

First, I would like to thank you very much for your scientific work which is interesting as tried to evaluate a very interesting field. But from the other side I believe that’s necessary to improve some points:

  1. 2.2 Single lap specimen. Have you considered to use another configuration of FBG’s instead of 3 FBG’s? For example, 2 in different positions or other number? Using different FBG’s diameter? And if yes what are the conclusions? Are the same?
  2. 2.2 Single lap specimen. Have you performed tests to evaluate that FBG’s are well positioned and aligned? Especially in order to explain not symmetrical results. What is the influence not well aligned in the next results?
  3. 2.3 Mechanical testing. Please provide testing standard which you are using. Have you performed tensile tests without FPG’s to evaluate mechanical performance degradation to presence of these?
  4. 2.3 Mechanical testing. What is the influence of interacting procedure to the mechanical performance of the specimen?
  5. 2.3 Mechanical testing. What is the influence of the small deflection of the specimen during the test? Have you evaluated this using for example an DIC equipment?
  6. 2.3 Mechanical testing. Have you evaluated presence of FBG’s using FEM simulation? Specially to identify not symmetrical failure distribution which next described.
  7. 2.3 Mechanical testing. Have you considered standard deviation of the adhesive performance at your received data? What is the influnce in the results?
  8. 3 Results. 3.1 Damage monitoring. 3.1.1 Spectrum evolution under testing. Have you checked each FBG in different position and load value after tests to identify that data are correct about wavelength? Can we evaluate same behavior in each specimen? What is the difference in results?
  9. 3 Results. 3.1 Damage monitoring. 3.1.1 Spectrum evolution under testing. Have you evaluated FBG’s behavior with a FEM model? Or using another scientific tool?
  10. 3 Results. 3.1 Damage monitoring. 3.1.1 Spectrum evolution under testing. Have you evaluated FBG’s data with after testing fracture surface evaluation? What kind of type of failure have you observed? Cohesive or adhesive? What is the influence of FBG’s presence on that? There is a possibility to observe this behavior due to a small deflection of the specimen or due to different adhesive performance?
  11. 3 Results. 3.1 Damage monitoring. 3.1.2 Damage monitoring using unloaded spectra. Please provide wavelengths for 400 N for all FBG’s to evaluate behavior after unloading.
  12. 3 Results. 3.1 Damage monitoring. 3.1.3 Quantitive comparison of unload spectra with the reference spectrum. How can you explain that the V for the FBGL and FBGR is not the same as probably expected? Have you evaluated optically after test?
  13. 3 Results. 3.1 Damage monitoring. 3.1.3 Quantitive comparison of unload spectra with the reference spectrum. Please explain FBGM behavior between 80 -100 failure load before unloading.
  14. 3 Results. 3.2 Damage monitoring during cyclic fatigue loading. 3.2.1 Virgin specimens fatigue testing. Please explain better different behavior between FGBL with FGBR/FGBM. Have you noticed same behavior in all specimens?
  15. 3 Results. 3.2 Damage monitoring during cyclic fatigue loading. 3.2.2 Fatigue testing of specimens pre-tensioned to before damage emergence. Please explain why FBGL power intensity varies from -45dBm to about -27dBm and FBGR varies from -55dBm to about -27dBm?
  16. 3 Results. 3.2 Damage monitoring during cyclic fatigue loading. 3.2.2 Fatigue testing of specimens pre-tensioned to before damage emergence. Please explain why V for FBGR is not identical with FBGL.
  17. 3 Results. 3.2 Damage monitoring during cyclic fatigue loading. 3.2.2 Fatigue testing of specimens pre-tensioned to before damage emergence. Please explain why is missing 100.000 cycles for FBGL and FBGM.
  18. 3 Results. 3.2 Damage monitoring during cyclic fatigue loading. 3.2.3 Fatigue testing of specimens pre-tensioned to damage emergence. Please explain Please explain why FBGL power intensity varies from -70dBm to about -45dBm and FBGR varies from -90dBm to about -45dBm are also aren’t identical? What about wavelength of maximum power intensity? Why is not the same? This behavior can we observe in all specimens?
  19. 3 Results. 3.2 Damage monitoring during cyclic fatigue loading. 3.2.3 Fatigue testing of specimens pre-tensioned to damage emergence. Why the shape of the graph is also identical? Have you checked after test to evaluate other critical points like fracture surface? Have you performed any optical inspection?

Reviewer 3 Report

The researchers have done excellent experimental work that shows that the FBG spectrum should change when it is chirped. This generally confirms the theoretical predictions.

In this article, I lacked a chapter of mathematical modeling that would describe the change in the spectral shape of the FBG depending on the chirp of its period.

The authors showed that, upon deformation of a fiber Bragg grating, additional peaks may appear in it, but they could not explain the nature of this phenomenon. However, this fact is obvious. Under uneven stretching of a homogeneous fiber Bragg grating, if one part of the fiber Bragg grating is stretched and the other part is in rest, it will split into two gratings with two different periods. Consequently, there will be two reflection peaks in the spectrum. The spectral width of these peaks will increase as the two gratings become shorter. This is very clearly seen in Figure 2 at a load of 6600 Newtons. In the manuscript work, I did not find any description of it. It seems to me that I would draw the following conclusion that with an increase in the load to 5200 Newtons, we can assume that the fiber Bragg gratings retain their shape, and with a further increase, its shape degrades.

The authors introduced parameter V, which describes the difference between the two spectra. At the same time, this parameter is dimensional, which is not very good, since it does not allow comparisons for experiments carried out on different spectrum analyzers. How can one understand how much one FBG was deformed relative to another if the data were measured on different spectrum analyzers? In my opinion, the authors should revise their approach to determining parameter V, so as to make it a dimensionless quantity. It would be nice if parameter V was built in relation to the increase in load, and not to fatigue.

The authors did not make any comments on the effect of temperature on the shift of the central wavelength of the fiber Bragg grating. There is no mention of how the effect of temperature can be compensated for on parameter V. This is a big omission.

In addition, the authors failed to show exactly how the unevenness of the grating period will change depending on the load. It was only demonstrated that with uneven stretching of the grating, its spectrum will change, but how exactly this will happen is not clear.

The conclusions based on the results of the research seem to confirm the conducted research, at the same time, we would like them to contain specific recommendations for the construction of such measuring systems.

I was also confused by the fact that among the list of references, just over half of the articles were published in the last five years. That is, the authors refer to old publications, especially when referring to fiber Bragg gratings.

In addition, it seems to me doubtful that the method proposed by the authors can be widely applicable in the industry. Modern spectrum analyzers are quite expensive, and there can be a lot of adhesive control points, which will require a huge number of fiber Bragg sensors. Consequently, the method proposed by the authors can be applied in some individual exceptional projects.

At the same time, the method proposed by them can indeed have great potential if we go from spectral measurement methods requiring spectrum analyzers to microwave photonic methods, say, using fiber Bragg gratings with phase shifts.

Round 2

Reviewer 3 Report

I am satisfied, the authors have made all the edits to the comments that I made.